# Nanoscale Engineering of Inorganic Composite Scintillation Materials

**DOI:** 10.3390/ma14174889

**Published:** 2021-08-27

**Authors:** Mikhail Korzhik, Andrei Fedorov, Georgy Dosovitskiy, Toyli Anniyev, Maxim Vasilyev, Valery Khabashesku

**Affiliations:** 1National Research Center Kurchatov Institute, 123098 Moscow, Russia; 2Radiation Instruments and New Components LLC, 220600 Minsk, Belarus; riincllc@yahoo.com; 3Baker Hughes, Houston, TX 77013, USA; toyli.anniyev@bakerhughes.com (T.A.); maxim.vasilyev@bakerhughes.com (M.V.); 4Department of Materials Science and Nanoengineering, Rice University, Houston, TX 77005, USA

**Keywords:** compositional disorder, ceramics cerium, glass ceramics, inorganic composites, nano-engineering, scintillators

## Abstract

This review article considers the latest developments in the field of inorganic scintillation materials. Modern trends in the improvement of inorganic scintillation materials are based on engineering their features at the nanoscale level. The essential challenges to the fundamental steps of the technology of inorganic glass, glass ceramics, and ceramic scintillation materials are discussed. The advantage of co-precipitation over the solid-state synthesis of the raw material compositions, particularly those which include high vapor components is described. Methods to improve the scintillation parameters of the glass to the level of single crystals are considered. The move to crystalline systems with the compositional disorder to improve their scintillation properties is justified both theoretically and practically. A benefit of the implementation of the discussed matters into the technology of well-known glass and crystalline scintillation materials is demonstrated.

## Highlights

The engineering of different aspects of the technology of the inorganic scintillation materials at the nano-level is a basis for their future improvement.

Glass scintillators have the potential to be improved to the level of luminosity of crystalline scintillation materials.

Compositional disorder is a powerful tool for the future improvement of the scintillation properties of inorganic crystalline materials.

## 1. Introduction

Nowadays, inorganic scintillation materials play an important role in instruments for measuring ionizing radiation. To date, more than two hundred scintillation materials have been discovered [1]. This took more than a century. In the 1940s, when nuclear instrumentation demanded new techniques to detect ionizing radiation, a set of the alkali-halide crystalline materials, particularly NaI(Tl) and CsI(Tl), was discovered. In addition to Tl^1+^, Eu^2+^ was recognized to be a very promising activator in alkali-halides. The discovery of the scintillation properties of a variety of oxide materials opened a wide perspective for heavy crystals, particularly doped with Ce^3+^ or Pr^3+^.

The needs of the experimental work in high energy physics demanded a new kind of scintillation materials tolerant to irradiation. Firstly, we note a forty-year story for the confirmation of the Standard Model describing the Universe, which required three generations of accelerators at the European laboratory for nuclear research (CERN Geneva, Switzerland): the proton synchrotron (PS), the super proton synchrotron (SPS), and the large hadron collider (LHC). Self-activated Bi_4_Ge_3_O_12_and PbWO_4_ were found to be the most promising candidates for electromagnetic calorimetry in the SPS and LHC experiments. The PbWO_4_ scintillator has played a crucial role in the discovery of properties of matter: the electromagnetic calorimeter of the Compact Muon Solenoid CMS collaboration at the LHC [2], made of 11 m^3^ of PbWO_4_ scintillation crystals, allowed for the triumphant discovery of a new boson predicted by Higgs [3].

Doped with Ce oxide, crystalline scintillators allow for the combination of a unique set of functional properties, which are called 3H: high stopping power to ionizing radiation, high light yield, and high time resolution. A good example is the lutetium oxyorthosilicate (Lu_2_SiO_5_-(Lu,Y)_2_SiO_5_) family [4,5,6], which is widely used for positron emission tomography (PET) manufacturing. The application of Lu_2_SiO_5_ type crystals made possible a new generation of time-of-flight PET scanners in single- and multi-modal combinations [7].

Traditionally, most of the applications are occupied by crystalline materials obtained by melt methods and, for some materials, in the form of ceramics and films. Crystal growth by the well-developed techniques invented by Chochralski, Bridgeman, Stokbarger and Stepanov makes it possible to obtain crystal blocks of a relatively simple rectangular or cylindrical shape. The creation of periodic structures or elements of complex shapes requires significant processing efforts. At the same time, new methods for producing bulk samples such as 3D printing [8] and the use of glass materials and their glass ceramics can significantly expand the range of products attractive to end-users. In such materials, which are essentially composites, a number of problems arise which can be solved through optimization at the nano-level. Particularly, these problems include the control of defects in both the region of the grain boundaries in crystals and the boundaries of the amorphous and crystalline phases in glass-ceramics. Another trend that has emerged in the last few years is the application of new advances in the theory of inorganic scintillation materials to optimize their properties for specific applications [9]. From this point of view, the production of nanostructured transparent glass-ceramics from a heavy glass based on rare-earth elements seems to be a promising way to reach properties close to those of single crystals. In crystals with compositional disordering at the level of a primitive cell, when the spatial symmetry of the crystal retained by the anionic sublattice remains the same, it is possible to significantly change the dynamics of the nonequilibrium carriers created by ionizing radiation, so the process of energy transfer becomes different from purely ordered materials. Thus, the step-by-step transition to a high entropy crystalline material promises advantages for the future improvement of the scintillation properties of currently existing scintillation materials, the technology of which is already well developed and implemented. This article considers the essential challenges to the engineering of the glass, glass ceramics, and ceramics scintillation materials, which can be clarified and resolved at the level of nanoengineering. These challenges include (i) the conservation of the stoichiometric composition of the material with high accuracy, (ii) control of the charge state of the activating ion in the material and grain boundaries, (iii) creation of the transport conditions for excitons in the glass, (iv) control of nonequilibrium carrier scattering in the materials to result in better luminosity and time resolution.

## 2. Conservation of the Stoichiometric Composition with a High Accuracy

The scintillation properties of the crystal are strongly related to the degree of perfection of the inorganic stoichiometric composition. In synthetic crystalline materials, where the growth rate of crystal mass is very high, the concentration of defects exceeds the thermodynamic limit by orders of magnitude. At the same time, the set of defects which are created during production strongly depends on the raw material preparation prior to melting.

The purity of the raw material has a crucial role. Uncontrolled impurities join the raw material at different stages of the technology of chemical preparation. From the point of view of stability under ionizing radiation, heterovalent impurities become of particular interest for inspection because their excess (or deficit) charge must be compensated by other point defects and, as a consequence, will inevitably generate another point defect. As a rule, heterovalent impurities form impurity-vacancy dipoles, and the electric neutrality of the lattice (charge compensation) is conditioned by their close association with the corresponding vacancy.

When the purity of the material is not a dominating factor in defect generation, point defects based on ion vacancies and interstitials and their associates [10] become the most influencing factor affecting the scintillation properties.

The point structure defects creation mechanisms in the crystalline and amorphous scintillation inorganic media at their production differ. In a glass and glass ceramic, the incorporation of modifying ions into the glass net, as a rule, does not positively affect their scintillation properties. However, prominent scintillation properties were obtained with the glass and glass ceramics obtained from binary (Li_2_O-2SiO_2_; BaO-2SiO_2_) and ternary (Li_2_O-(2−x)SiO_2_-xAl_2_O_3_; (1−x)BaO-xRE-2SiO_2_ (RE = Gd, Lu)) [11,12,13] compositions. The aluminum ion, up to some critical concentration of ~7 at.%, remains a net creating ion, similarly to a silicon ion, and occupies an oxygen tetrahedron. It does not introduce an additional glass structure modification. An introduction of rare earth (RE) ions in the glass was determined to be an effective technological way to reduce amount of unlinked oxygen ions in the glass [14]. Contrary to silicon and aluminum ions, RE ions are stabilized among their tetrahedra, and at the nanoscale level they promote a termination of the unbonded oxygen links. Due to this reason, barium-silicate glass, heavy loaded with RE, shows good resistance to ionizing radiation [15].

In multicomponent Al/Ga garnets, which became of particular interest in recent years [16,17], the production process of single crystal growth or ceramic sintering at high temperatures is complicated by the high evaporation rate of Ga oxide from the melt. Ga_2_O_3_ has a maximum vaporization rate 4–5 orders of magnitude higher than that of Al_2_O_3_ in the 1600–2000 °C temperature range [18]. Predominant Ga oxide evaporation makes it difficult to precisely control crystal composition.

Small variations of composition in Gd_3_Ga_3_Al_2_O_12_:Ce and related (Gd,Y)_3_(Ga,Al)_5_O_12_:Ce materials (of the order of several atomic percent) significantly influence their scintillation properties—light yield, kinetics, and afterglow—as has been shown in numerous studies, e.g. [19,20]. A possible solution was proposed in [21], where the advantage of the co-precipitation of the raw material for a solid-state synthesis being manifested. Co-precipitation allows us to achieve a high homogeneity of the component’s distribution in the product, which enhances phase formation at much lower temperatures than for solid state reaction (e.g., 1000–1200 °C instead of 1400–1600 °C). When incorporated into the garnet structure, Ga is expected to demonstrate a lower evaporation rate. Figure 1 shows SEM images of the Gd_3_Ga_3_Al_2_O_12_:Ce powder, co-precipitated and further sintered at 1200 °C, and of the same material obtained by solid-state synthesis at 1400 °C. The morphology was found to be identical in spite of the latter being produced at a temperature 200 °C higher. Pieces of the powder obtained from coprecipitated raw material already consist of small crystallites with the garnet habitus having a typical size of less than one micron. Particles obtained by solid-state synthesis have a similar look as well, however, they were obtained at a high temperature. Moreover, contrary to a solid-state synthesis, the co-precipitation approach helps to reduce the creation of undesirable phases (other than garnets) during sintering.

## 3. Control of the Charge State of Rare-Earth Activating Ions in the Inorganic Scintillation Materials

The general trend of today is to search for fast scintillators. Combining fast and bright scintillation with a high stopping power to various kinds of ionizing radiation provides an opportunity for good detection efficiency and high count rate. Besides cross-luminescence and strongly quenched luminescence in self-activated materials [1], fast decaying scintillation in inorganic compounds can be obtained when they are activated by rare-earth ions having the radiative transition 4f^n−1^5d → f^n^, which are allowed both on spin and on parity. The isovalent substitution of matrix ions by Ce^3+^ ion in many inorganic compounds results in a high quantum yield and fast luminescence. Ce^3+^ ions have a rather simple structure of energy levels according to the data from [22]. The basic 4f configuration of the Ce^3+^ ion is the ^2^F_7/2,5/2_ spin-orbit doublet with an energy gap of ~ 2400 cm^−1^. This difference remains practically the same in inorganic compounds due to the strong effect of the spin-orbit interaction on f-states, as compared with a Stark effect from the crystalline field of ligands. On the contrary, the 5d-orbital, which is split into 5 components ^5^d_1–5_, is strongly influenced by the symmetry and dimensions of the anionic polyhedra surrounding Ce^3+^ dopant ions. This provides a way to engineer the positions of the electron states of the Ce^3+^ ions relative to the conduction and forbidden zones in the crystalline material [23]. In the case of glass, where the Ce^3+^ activator ion has a spectrum of the positions for localization, and the crystalline field is formed by the ligands of the chaotically oriented net creating polyhedra such as tetrahedra in the silica glass, the engineering tools are quite different from those which are applied in crystalline materials. Here, in practice, it is difficult to manipulate the disposition of the electronic levels in the band gap to reach a satisfactory transfer of matrix electronic excitations to Ce^3+^ ions; therefore, one high priority is the high concentration of the activator in the material and its dominating trivalent charge state.

### 3.1. Glass Scintillators

One possible way to achieve the complete stabilization of Ce ions in the trivalent state at a high concentration in a glass is the addition of reducing agents to the glass charge, which become oxidized upon the melting of the glass, providing the reductive atmosphere. Their type and optimal amounts are determined by the specific compositions of the glass [24], as well as by other factors mentioned below. Oxide glasses and glass-ceramics, made from binary and ternary stochiometric composition metal salts and SiO_2_, have been found to be good matrices for the development of scintillation materials doped with Ce ions. Among light glasses, only Li-based compounds [13,25,26] allow for the creation of glass scintillators with a relatively high light yield, which is nevertheless a few times lower than the typical light yields of single-crystal scintillation materials [1]. Nevertheless, they allow for a relatively high concentration of Ce^3+^ ions and are successfully used to detect thermal neutrons when the glass is enriched with ^6^Li ions [27,28].

The equilibrium structure of a melt is determined by its composition and temperature, and partially by the atmosphere of the glass melting furnace. The reducing agent reacts during the process of the glass melting; therefore, its optimal amount varies with reaction rate and process time and is influenced by technological factors such as the size and shape of the crucible, peculiarities of the furnace construction, etc. At the same time, the equilibrium ratio of Ce^3+^/Ce^4+^ in glasses of different compositions can be established within 10–60 h [29]. This justifies the application of another method to influence the Ce^3+^/Ce^4+^ ratio: the use of the pre-synthesized compounds containing Ce^3+^.

Cerium silicates were chosen as the pre-synthesized components. Three compounds are formed as the result of reaction of CeO_2_ and SiO_2_ in strongly reducing conditions (dry hydrogen atmosphere): Ce_2_O_3_ ∙ SiO_2_ (Ce_2_SiO_5_), 2Ce_2_O_3_ ∙ 3SiO_2_ (Ce_4_Si_3_O_12_), and Ce_2_O_3_ ∙ 2SiO_2_ (Ce_2_Si_2_O_7_), and they decompose at temperatures of 300–500 °C, 600–700 °C and 900 °C, respectively, when heated in air [30]. The most stable of them, cerium pyrosilicate (Ce_2_Si_2_O_7_), is reported to be stable in air at temperatures above 1600 °C. Mixed pyrosilicates of cerium with other lanthanides, which have a single oxidation state of 3^+^, show higher stability in air, e.g., (Ce_0.5_Nd_0.5_)_2_Si_2_O_7_ decomposes at 1300 °C only by 3%, and (Ce_0.25_Nd_0.75_)_2_Si_2_O_7_ and (Ce_0.5_La_0.5_)_2_Si_2_O_7_ are completely stable [31]. Moreover, gadolinium pyrosilicate activated by cerium is a scintillator with a sufficiently high light yield of 30,000–45,000 photons/MeV or higher [1]. Thus, it can be assumed that introduction of pyrosilicate blocks containing Ce^3+^ ions into a glass would contribute to an increased yield of scintillations. Variation of the set of rare-earth ions in a pyrosilicate compound allows us to achieve the best scintillation characteristics of glasses. Such materials demonstrate a scintillation yield 15–20% higher than that of materials obtained using CeO_2_ in a glass composition.

It is worth noting that lithium glass has low density and therefore is practically useless for the design of detecting elements for the detection of other types of ionizing radiation, particularly the γ-quanta of different energies. On contrary, a barium-silicate glass made from a stochiometric composition, such as BaO_2_-SiO_2_, allows for a high concentration of Ce^3+^ ions [11,12]. Furthermore, glass made on the basis of barium–RE silicate composition also demonstrates a capability to incorporate Ce^3+^ with a relatively large quantity. This type of glass is produced from the commercially available salts for the initial charge and can be manufactured in large quantities using standard technology in the glass industry. It provides relatively fast energy response, and although its light yield is lower than that of the most efficient scintillators, it exceeds by ten times the light yield of the PbWO_4_ scintillator which is nowadays the most widely used scintillating material in HEP experiments [32,33]. Adding RE ions into the glass composition may increase the density of the resulting glass to 5.3 g/cm^3^.

### 3.2. Ceramic Scintillators

One may expect that certain inherent features of ceramic scintillators based on complex oxide compounds would facilitate the cerium activator oxidation to the Ce^4+^ state. A significant difference between polycrystalline ceramics and single crystals is that ceramics contain grain boundaries separating different crystallites. Figure 2 shows images of GYAGG ceramics activated by Ce ions and their grain boundaries. It can be seen that the grains in the ceramics are oriented differently.

Cerium ion segregation to the grain boundaries in YAG:Ce and GdYAG:Ce ceramics was recently demonstrated [34,35]. The phenomenon has been related to the common reason for dopant partitioning in melt-grown crystals [36]. Photoluminescence enhancement was found to occur in the grain boundary region, whereas the influence of this process on scintillation is still unclear. At the same time, GYAGG:Ce-mixed garnet ceramic samples obtained by hot pressing show a scintillation yield exceeding that of a single crystal, up to 55,000 ph/MeV [37,38]. The authors of [39] recently confirmed Ce grain boundary segregation in GAGG:Ce ceramics, but according to the measured data, the fraction of such ions was estimated to be below 1% of the total Ce content. The photoluminescence at the grain boundaries was measured to be more intense compared to the bulk of the grains, as it takes place in YAG:Ce (Y_3_Al_5_O_12_:Ce) ceramics. The presence of the boundaries could bring positive aspects to the process of scintillation as well. Apparently, the migration of excitons across the grain boundaries is hindered and, therefore, they predominantly recombine within the same grain in which they were generated. This leads to an effective exciton scattering length reduction and, therefore, to a more efficient transfer of exciton energy to the Ce^3+^ activator centers, resulting in an increase in the scintillation yield of ceramics, as compared to the same single-crystal material.

**Figure 2 materials-14-04889-f002:**
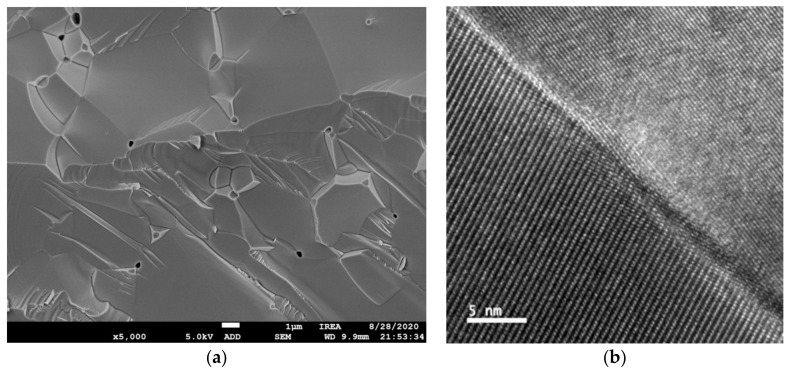
SEM image of (Gd,Y)_3_Al_2_Ga_3_O_12_:Ce ceramics, polished and thermally etched, in backscattered electrons mode showing grain structure (**a**) and intergrain boundary (**b**), after [39].

## 4. Creation of the Transport Conditions for Excitons in the Glass

Glass, being of a disordered structure, is not conditioned to provide long-range migration of excitons created in the matrices by ionizing radiation. A partial ordering of the glass by crystallization can mitigate this problem [11], but not resolve it completely. The incorporation of the functional capabilities for the transport of nonequilibrium carriers into the glass is one of the possibilities for providing effective delivery of electronic excitations to Ce^3+^ luminescence centers. In this view, one of the possible ways to increase the light yield of the scintillation in Ce-doped inorganic glass is introducing rare-earth ions into the inorganic matrix. Among those, on a base of ^6^P states, Gd (f^7^ electronic configuration) could introduce the subzone into the band gap when the concentration of the ions in the glass composition is sufficient to provide the effective migration of electronic excitations along the Gd subsystem. It is known that the ^8^S ground state of Gd^3+^ ions in oxide compounds is located in the valence band near its top [40]. Therefore, Gd^3+^ ions possess a capability for creation of Frenkel-type excitons, in which nonequilibrium electrons are linked to ^6^P and holes are linked to the ^8^S states.

As a result, the light yield of the glass can reach up to the 50% level of Bi_4_Si_3_O_12_ [1]. In parallel, the density can be increased to 4.2 g/cm^3^ or even higher, which opens the possibility of detecting low-energy γ-quanta with acceptable energy resolution. Nevertheless, further improvement to the scintillation light yield of the Gd-filled glass is connected with the managing of better resonance conditions between the Gd^3+^ subsystem and Ce^3+^ ions. Figure 3 shows the low-energy electronic levels of Gd^3+^ and Ce^3+^ ions in Ba-Gd-Si glass.

At a high concentration of Gd in the glass host, a significant part of the nonequilibrium carriers create Frenkel-type excitons located in the Gd^3+^ ions subnet, which has migration capability. The Gd subsystem may contribute to the population of the Ce^3+^ radiating level through the interaction of Frenkel exciton and the ^2^F_5/2_→5d_2_ transition, however, the resonance is quite poor. Moreover, the 5d_2_ levels of the set of Ce^3+^ created in the glass are located in a conduction band, so a delocalization of electrons occurs, allowing the nonradiative recombination of nonequilibrium carriers. Holes can be quickly delocalized to the top of the valence band from thr ^8^S state [40]. The transfer process, which involves the resonance between ^2^F_5/2_→5d_1_ and ^6^P→VB_top_ (top of the valence band), is poor as well. This resembles the situation in Gd_2_SiO_5_:Ce, in which the localization is small in the crystalline field in the 7(O) and 8(O) positions of Ce ions, the resonance conditions between Frenkel excitons and ^2^F_5/2_→5d_2_ are low, and consequently the light yield is lower than 10,000 ph/MeV [1]. An increase in the crystalline field strength in the Ce^3+^ stabilization positions in the glass becomes necessary in future glass modifications. This was demonstrated to be a well-working approach on the transfer from Gd-silicate to Gd-garnet type crystals, in which the light yield has increased 3 times and even more [41]. A possible solution is a search for glass crystallization capabilities which on one hand will allow a multi-seed glass crystallization at volume and, on the other hand will not deteriorate the optical transparency of the glass.

## 5. Control of Nonequilibrium Carrier Scattering in the Conduction Band of Disordered Materials

Improvement of the technology of already widely used scintillation materials gives a certain chance for their better properties but does not provide breakthrough changes in their mass production parameters. Thus, selected grade specimens of materials appear, which are significantly more expensive than those that are produced under ordinary technological conditions. A global trend in the production of scintillators is a transition to materials with a more complex composition, both in the cationic and anionic sublattices.

Pioneering experiments on improving the scintillation properties by growing a mixed perovskite-type Lu_x_Y_1−x_AlO_3_:Ce instead of LuAlO_3_:Ce and oxyorthosilicate type (Lu_x_Y_1−x_)_2_SiO_5_:Ce instead of Lu_2_SiO_5_:Ce were performed more than two decades ago [42,43,44]. Impressive capabilities for improving the scintillation properties in garnet-type mixed crystals by engineering the composition were demonstrated on (Lu,Gd)_3_(GaAl)_5_O_12_ of various compositions [45,46]. Further, this activity was transformed into a combinatorial search for best composition of Ce-doped multiionic garnet compounds containing Y, Gd, Ga, and Al using the micro-pulling down method as a very flexible and cost-effective crystal growth technique to obtain small single crystal samples for express research [47]. Based on this study, large diameter Gd_3_Al_2_Ga_3_O_12_:Ce (GAGG:Ce) single crystals were grown using the Czochralski method, which exhibited similar or even higher LY of 46,000 ph/MeV [48,49,50]. In the last decade, the search for new high light-yield scintillation materials, albeit not necessarily fast, produced a number of examples of successful studies. Renewed interest for SrI_2_:Eu has brought about a technology upgrade [51,52], but the most impressive results we obtained with mixed halides [53]. Besides improvement of the scintillation properties, a mixing of halide anions allows for the elimination of the problem of halide material hygroscopicity [54].

The creation of disorder in a cationic sublattice at the conservation of the charge balance by a mixture of isovalent ions in the crystalline compound does not introduce a similar disorder in the anionic subsystem, which keeps on track in an ordered state. It keeps a set of the phonons, but modifies them following the local change of the cation’s nuclei, therefore keeping good capabilities for nonequilibrium carrier migration along the crystal lattice. Moreover, the random alternation of different cations in the crystal lattice leads to a phenomenon that causes the shortening of the scattering tracks of photoelectrons formed in the scintillation material due to the photo-absorption of gamma radiation. Consider a widely used (Lu_1−x_Y_x_)_2_SiO_5_:Ce (LYSO) scintillator, which in many applications has replaced Lu_2_SiO_5_:Ce (LSO). Crystals of LSO and YSO have a monoclinic lattice which contains 64 atoms. Of these, 16 are lutetium and yttrium ions, respectively. Disordered LYSO is a lattice of the same structure with randomly filled 6(O) and 7(O) sites by Lu and Y ions. In view of the significant similarity between the band structures of these crystals [55], their lattice constants, and relative positions of the sites, it can be assumed that the filling of the sites is statistically independent. The difference in the bandgap is small, ΔE_g_~80 meV, and when replacing Lu with Y; it is the edge of the conduction band that is formed by the s-states of the valence electrons of these ions that is shifted.

Fluctuations of the conduction band edge can be simulated by the pseudopotential method [40], which considers only the short-range order, while the pseudopotential profile itself has a lattice symmetry. The results of modeling the fluctuations of the conduction band edge in the Lu_1.6_Y_0.4_SiO_5_ crystal are shown in Figure 4.

The average scattering length of electrons in the conduction band on these fluctuations can be calculated using the scattering cross section in the Born approximation for energies. Although the amplitude of the fluctuations of the pseudopotential is not large, it should be noted that the fluctuations themselves can have a significant effect on the electronic states near the edge of the band; that is, at the stages of the formation of exciton states, shortening the length of their diffusion and the capture of the nonequilibrium carriers. Typically, LYSO (Lu_2_SiO_5_:Ce) has a light yield 20–30% over that of LSO, and its coincidence time resolution is measured to be less than 100 ps.

## 6. Conclusions

We described key technological challenges and the possible resolution methods in the development and production of oxide inorganic scintillation materials. In view of the trend to increase the complexity of the materials, the crucial role of the procedures for the preparation of raw material is underlined. Utilization of the starting composition produced by co-precipitation or use of the salts of the dopants with the predetermined valent state was found to be a prospective way to improve the material properties.

The effect of the Gd subnet in inorganic compounds was considered. Tuning of the resonance conditions between Frenkel-type excitons, which are localized at the subnet, and electronic transitions of activating ions is a key tool to arrange for better properties in both crystalline materials and glasses.

In this work, we did not touch upon the direction, that has just begun to develop, on the formation of the profile of the crystal field in the positions of the localization of activator ions, especially in the crystalline inorganic systems used for scintillation materials. Compositional disorder in multi-cationic compounds inevitably leads to a multiplicity of positions of localization of the impurity centers in crystals. As a result, a significant difference arises in the spectral-luminescent properties, and luminescence centers with very different emission kinetics appear. For example, within the luminescence band of Ce^3+^ ions in quaternary garnets, the decay time constant of photoluminescence can vary as much as 30%, which is not typical for binary systems. Since the shortening of the scintillation kinetics while maintaining a high light yield, is the key for further progress in the time resolution of scintillation detectors, the next step will undoubtedly be the search for technological conditions for the preferential localization of such fast luminescence centers in disordered crystals. We see this as a promising area of application for nanoengineering.

## Figures and Tables

**Figure 1 materials-14-04889-f001:**
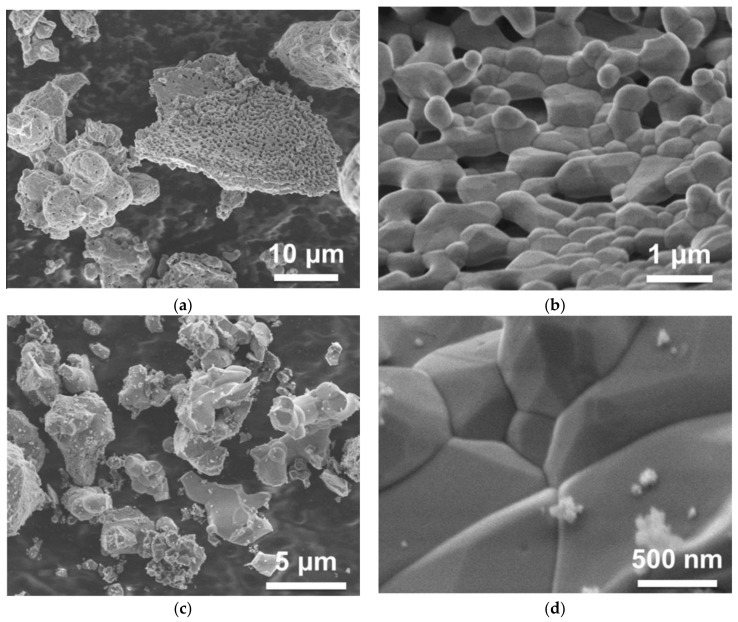
SEM images of co-precipitated Gd_3_Ga_3_Al_2_O_12_:Ce powder after thermal treatment at 1200 °C (**a**,**b**), and powder obtained by solid state synthesis at 1400 °C (**c**,**d**).

**Figure 3 materials-14-04889-f003:**
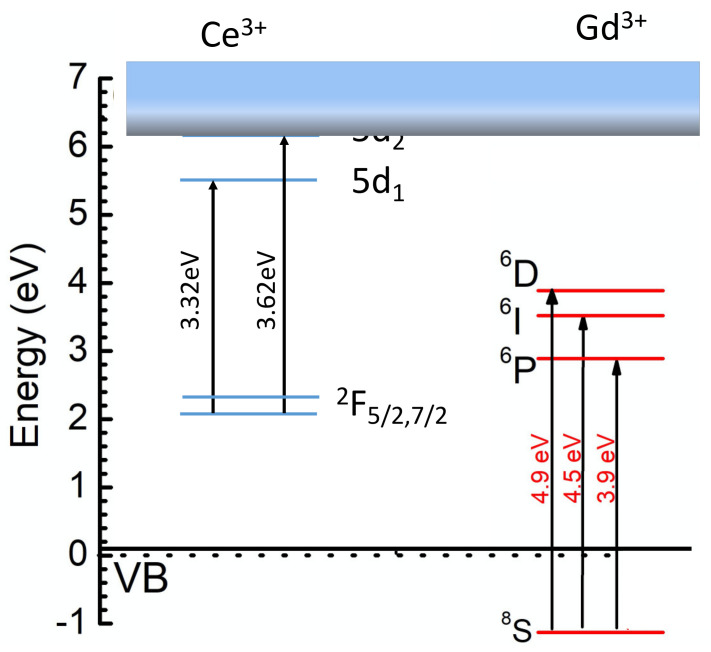
Sketch of the low-energy electronic levels of Gd^3+^ and Ce^3+^ ions in Ba-Gd-Si glass.

**Figure 4 materials-14-04889-f004:**
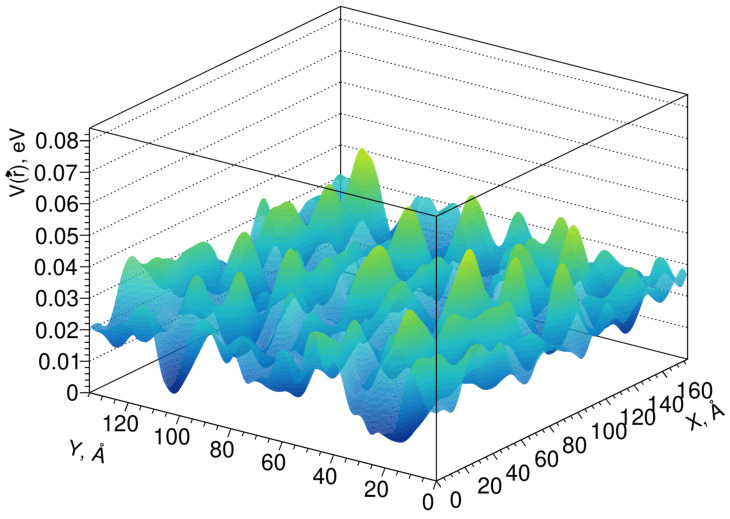
Profile of fluctuations of the conduction band edge in a Lu_1.6_Y_0.4_SiO_5_ crystal.

## Data Availability

All the data is available within the manuscript.

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
