# Peer review of "Nanoscale Engineering of Inorganic Composite Scintillation Materials"

_materials, 2021, doi:10.3390/ma14174889_

Round 1
Reviewer 1 Report
In this article, the authors considered the ways to improve the scintillation parameters of the glass to the level of single crystals.
They pointed out the crucial role of the procedure for the preparation of the raw material and found that utilization of the starting composition produced by co-precipitation or use of the salts of the dopants with the predetermined valent state is a prospective way for the materials properties improvement.
I think this article is interesting for the both of fundamental and engineering aspects and has worth to be published in this journal.
Only some misprints stated below should be corrected before publication.
L.59 stochiometric → stoichiometric
LL.86-87, L.184 To represent binary or ternary system, usurely not use the symbol *, but - such as A - B or A - B - C.
L.141 bandgap → band gap
L.406 The font of "40" is large and bold.
Author Response
Reviewer 1 |
||
|
Reviewer comment |
Authors response |
1 |
L.59 stochiometric → stoichiometric |
Corrected |
2 |
LL.86-87, L.184 To represent binary or ternary system, usurely not use the symbol *, but - such as A - B or A - B - C. |
Corrected |
3 |
L.141 bandgap → band gap |
Corrected |
4 |
L.406 The font of "40" is large and bold. |
Corrected |

Reviewer 2 Report
Dear Sir,
The article is informative however it can be improved in some areas as follows:
In the abstract
- It is recommended to begin this part with general sentence about the main topic (scintillation materials) to grasp the reader’s attention.
- Authors can provide some promising findings and state of art in this part to attract the reader.
- keywords should be arranged alphabetically.
Introduction section:
- It is recommended to divide the manuscript into IMRAD structure (introduction, methodology, result and discussion and conclusion). To avoid any confusion if the research neither article or a review!!
In the other sections:
- Schematic diagram for the experimental part could be useful for the reader
- All chemicals being used in the study should be addressed in this section including where they are purchased or obtained.
- State the model, the company and country of origin of any instrument used in the research.
- Could the author give more information about the interpretation of images in Fig 1 and 2.
- Future work should be mentioned at the end of the conclusion
References
- References should be updated. The most updated reference was the year of 2015. Since this topic is important, there must be many recent references to be cited.
- FORMAT the references correctly according to the author’s guide such as ref. 22
Regards
Author Response
Reviewer 2 |
||
|
Reviewer comment |
Authors response |
1 |
In the abstract
|
Done. |
2 |
Introduction section:
|
We agree with the Reviewer that IMRAD structuring of the article is quite useful for original research articles dealing with consideration of one-two aspect of the problem. Here we touched a few directions and tried to show driving forces in the progress of the specific materials. We prefer to keep this presentation of the material, but to be closer to the Review format have significantly added the literature. |
3 |
In the other sections:
|
Schematic diagrams, description of the methods for the samples production, specification of the chemicals, instruments for research are denoted in the cited literature. If the Reviewer insists, we can manage a special chapter “Instruments for scintillation materials research”
More information is included
Conclusion updated accordingly |
4 |
References
|
References are updated and corrected |

Reviewer 3 Report
Your manuscript entitled “Nanoscale engineering of inorganic composite scintillation materials” is interesting for a publication in Materials. However, the title seems too general for the short content of the manuscript. The topic is very short in the Garnet crystals. Thus, my suggestion is to include “Garnet” in the title as some crystals like elpasolites may follow the same methods. Moreover, they are some recommendations prior the acceptance of this manuscript in Materials:
- For me, it is hard to follow the story of this manuscript. It is a review or an original article? From figure 1 to figure 4, the crystals are always different. I suggest if this is a review, it is better to present a table of comparisons among those scintillators with their performances and properties as ceramics and crystals.
- Page 2, lines 49-53: “it is possible to significantly change the dynamics of nonequilibrium carriers created by ionizing radiation, so the process of energy transfer becomes different from purely ordered materials.”. Please mention some examples of crystal lattices beside Garnet that their dynamics of nonequilibrium carriers can be changed. How about elpasolites or perovskites. I think that it is nice to discuss the possibility to extend this work with different structures.
- In Figure 2, you present the SEM images of the ceramics. However, for me, it is not clear what are a) and b). I think that it is important to present the distribution of Ce ions through EDX.
- Page 4, lines 156-157: “to detect thermal neutrons when the glass is enriched with 6Li ions”. Please cite some articles related to the similar structures as garnet (elpasolites) for these works, e.g., van Eijk et al., NIM A, 529, 1-3, 2004 and Birowosuto et al., J. Appl. Phys. 101, 066107, 2007.
- In Figure 4, you only put fluctuations of the conduction band edges in LYSO. Since in the previous figures, you mentioned other crystals, it seems that it will be better if you present all fluctuation profiles, e.g., GYAGG, in the same figure.
Author Response
Reviewer 3 |
||
|
Reviewer comment |
Authors response |
1 |
Your manuscript entitled “Nanoscale engineering of inorganic composite scintillation materials” is interesting for a publication in Materials. However, the title seems too general for the short content of the manuscript. The topic is very short in the Garnet crystals. Thus, my suggestion is to include “Garnet” in the title as some crystals like elpasolites may follow the same methods. Moreover, they are some recommendations prior the acceptance of this manuscript in Materials:
|
The cited literature was updated. We included references dedicated to mixed garnets, perovskites, and halides. So, we prefer to keep the title to be general. |
2 |
1.For me, it is hard to follow the story of this manuscript. It is a review or an original article? From figure 1 to figure 4, the crystals are always different. I suggest if this is a review, it is better to present a table of comparisons among those scintillators with their performances and properties as ceramics and crystals.
|
A general table with inorganic scintillation materials is published in ref [1]. We cited this book and rewritten an introduction accordingly. |
3 |
2.Page 2, lines 49-53: “it is possible to significantly change the dynamics of nonequilibrium carriers created by ionizing radiation, so the process of energy transfer becomes different from purely ordered materials.”. Please mention some examples of crystal lattices beside Garnet that their dynamics of nonequilibrium carriers can be changed. How about elpasolites or perovskites. I think that it is nice to discuss the possibility to extend this work with different structures.
|
We have described this in a wider context in paragraph 5. |
4 |
3.In Figure 2, you present the SEM images of the ceramics. However, for me, it is not clear what are a) and b). I think that it is important to present the distribution of Ce ions through EDX.
|
The EDX scan results of the boundary are shown in the reference [39]. We quoted this article. |
5 |
4.Page 4, lines 156-157: “to detect thermal neutrons when the glass is enriched with 6Li ions”. Please cite some articles related to the similar structures as garnet (elpasolites) for these works, e.g., van Eijk et al., NIM A, 529, 1-3, 2004 and Birowosuto et al., J. Appl. Phys. 101, 066107, 2007.
|
Included |
6 |
5.In Figure 4, you only put fluctuations of the conduction band edges in LYSO. Since in the previous figures, you mentioned other crystals, it seems that it will be better if you present all fluctuation profiles, e.g., GYAGG, in the same figure.
|
The simulation of the conduction band bottom fluctuations is a hard task. One way to get an impression of the complexity is when following progress in this field. Ref. [9] reported the peculiarities of the conduction band bottom modulation for two-ion systems. In two years, we found a possibility to describe correctly random modulation in a three-cationic system, namely LYSO. The simulation on four cationic GYAGG having three positions for cations stabilization is still in progress and as it seems to us will require a deep diving into the theory. Therefore, these results will be the subject of the next publications. As LYSO is described in the article, we decided to include this figure to provide a good illustration of the effect discussed. |

Round 2
Reviewer 3 Report
Thanks for the revision. I recommend to publish as it is